# Reproduction and Fertility of Buffaloes in Nepal

**DOI:** 10.3390/ani13010070

**Published:** 2022-12-24

**Authors:** Bhuminand Devkota, Shatrughan Shah, Gokarna Gautam

**Affiliations:** Department of Theriogenology, Faculty of Animal Science, Veterinary Science and Fisheries, Agriculture and Forestry University, Chitwan 44209, Nepal

**Keywords:** water buffalo, anestrus, nutritional status, seasonality, fertility management

## Abstract

**Simple Summary:**

Buffalo is the major livestock commodity in Nepal, contributing more than half of the total milk and more than one-third of the meat production in the country. One of the major constraints of buffalo production in Nepal is the low productive efficiency, due mainly to compromised fertility, characterized by delayed puberty, silent estrus, anestrus and seasonal breeding patterns. Poor management, reflected by endoparasitic infection and a low nutritional status, is found to be associated with anestrus and its treatment response in buffaloes. Recently, improved management, combined with timed artificial insemination techniques, has been adopted, which has improved the pregnancy outcomes. This review highlights the reproduction and fertility status of Nepalese buffaloes, the factors influencing fertility and the techniques that enhance the reproductive efficiency of buffaloes in Nepal.

**Abstract:**

Water buffalo (*Bubalus bubalis*) in Nepal contributes 57% of the total milk and 36% of the total meat production in the country. The productive efficiency of Nepalese buffaloes is quite low, due mainly to subfertility and infertility. Delayed puberty and prolonged inter-calving intervals, attributed mainly by anestrus due to silent cyclicity and ovarian acyclicity, are the major forms of infertility in Nepalese buffaloes. Moreover, buffaloes in Nepal show a distinct seasonal breeding pattern, with July to December as the active breeding season, and with April to June and January to March as the low and transitional breeding seasons, respectively. Endoparasitic infection and poor nutritional status, which are more severe during the low season, are found to be the major factors causing anestrus and compromising its treatment response in buffaloes. Various hormonal protocols for timed artificial insemination (TAI) have been attempted, with a varying pregnancy outcome. Recently, an integrated technique including anthelmintic treatment, nutritional supplementation and hormone-based fertility management programs for TAI has been developed and implemented successfully. A wider adoption of this technique as a package of practices could be key to improving the reproductive efficiency of buffaloes in Nepal.

## 1. Introduction

In a worldwide context, Nepal has the fourth-highest population of buffaloes (*Bubalus bubalis*) [1], with 5,159,931 heads in the year 2021 [2]. The agricultural land in Nepal is divided into three agro-ecological zones from north to south: the mountains, hills and Terai, or the plain. Buffaloes are raised across all agro-ecological zones of the country, under farming systems that range from large-scale and semi-intensive, with herd sizes of more than 50 animals, to small-scale, intensive systems in which farmers keep 1 to 5 animals, although the large-scale farms are very few and medium and small-scale farms predominate in the country. Buffalo production contributes 57.23% of the total milk and 36.13% of the total meat produced in the country [2]. The majority of livestock keepers in the country, mostly smallholders, derive their livelihoods from buffalo production. In addition to milk and meat production, the buffalo also makes a significant contribution to the farm economy through draught power and farm yard manure [3]. In Nepal, male buffaloes, especially on the flat plains in the south, also contribute to draught power, mainly in land preparation and the transportation of farm products in smallholder farms. However, female buffaloes are not used for draught power.

Almost all buffaloes in the country are of the riverine type, with the exception of a small population of swamp buffaloes, limited in wild form, in Koshi Tappu Wildlife Reserves [4]. It is estimated that 65% of the buffaloes in Nepal are indigenous [5], and they include the Lime, Parkote and Gaddi breeds [6]. These breeds are distributed mainly in the mountains and hills. The former two breeds are distributed from the eastern to western region and the third type in the far-western part of the country. On the other hand, there are the Terai buffalo in Eastern Terai, which was recently identified as the fourth indigenous breed [7], and exotic Indian Murrah or their crosses with indigenous breeds are distributed throughout Terai. Murrah buffaloes and their crosses comprise the remaining 35% of the national buffalo population [5].

The main product obtained through buffalo farming in Nepal is milk. However, the milk productivity of indigenous breeds of buffaloes is quite low, averaging 2.85 L per animal per day. This results in an estimated 869.3 L per animal in a 305-day lactation period [2]. The Murrah buffalo in the same system produces 1500 L of milk per lactation period [8]. In many areas, buffalo meat is preferred for some traditional cuisines, resulting in a high rate of slaughter of indigenous breeds such as the Lime. The rapid loss of indigenous breeds due to a slower rate of reproduction has highlighted the need to conserve the different breeds [9]. 

In the past few decades, the Nepalese government, through the Department of Livestock Services and the Nepal Agricultural Research Council, has implemented various programs to promote the productivity of buffaloes in different farming systems. These programs include the buffalo genetic improvement program, community buffalo bull distribution program, artificial insemination and forage missions, buffalo conservation, nutrition programs for newly calved buffaloes and male buffalo fattening for meat production in many parts of the country [8]. The overall objectives of the programs are to enhance buffalo productivity through improved nutrition and genetic improvement through the cross-breeding of local breeds using Murrah bulls or their semen. However, investments in monitoring anticipated change and adapting technologies to support genetic gains over generations are limited. The anticipated impacts in terms of enhancing the nutritional security, incomes and livelihoods of smallholder farmers have not been fully realized.

The productive efficiency of Nepalese buffaloes is quite low, due mainly to subfertility and infertility. Delayed puberty and prolonged inter-calving intervals, attributed mainly by anestrus due to silent cyclicity and ovarian acyclicity, are the major forms of infertility in Nepalese buffaloes. Moreover, buffaloes in Nepal show a distinct seasonal breeding pattern [10,11]. There is large potential to increase the productivity of buffaloes through the improvement of their reproductive efficiency. The aim of the present review is to highlight the reproduction and fertility status of Nepalese buffaloes, the factors influencing fertility and the techniques to enhance the reproductive efficiency of Nepalese buffaloes.

## 2. Reproductive Performance

Nepalese buffaloes have poor reproductive performance, characterized by a delayed age of first calving and prolonged inter-calving interval [12,13,14,15]. As shown in Table 1, the age at first service of indigenous buffaloes is around four years, with the exception of the Terai breed, in which it is nearly three years. As a result, the age at first calving is almost five years, with the exception of the Terai breed, in which it is around 44 months. It is interesting that the gestational length is longer in the Gaddi breed as compared to that in other indigenous breeds of buffalo. Similarly, the calving to conception interval is also quite long: it is around 6 months in the case of Lime and Parkote and even longer (i.e., almost a year) in Gaddi. As a consequence, the calving interval ranges from 20 to 23 months depending on the breed. On the other hand, in Murrah cross-breed buffaloes, the age at first conception, the calving to first estrus interval and the calving interval were reported as 2.9 ± 0.7 years, 3.5 ± 3.6 months and 14.0 ± 3.4 months, respectively [16], which indicates the better reproductive parameters in these buffaloes.

## 3. Infertility

Besides the delayed age at first calving and prolonged inter-calving interval, the productive efficiency of Nepalese buffaloes is hampered mainly due to subfertility and infertility, characterized by a silent estrus, anestrus, breeding seasonality and repeat breeding [12,13,14,15]. A previous study showed that 91.2% of infertility cases in buffalo were due to anestrus and the remaining 8.8% cases due to repeat breeding. Within the anestrous group, 54.8% buffaloes showed true anestrus characterized by ovarian acyclicity, and 45.2% showed silent estrus having a major structure of either dominant follicles (DF) or corpus luteum (CL) [15]. Another study revealed that 33.3% of the anestrous parous buffaloes had silent ovulation, while 61.4% showed true anestrus with ovarian acyclicity, and 18.9% of the buffalo heifers had silent ovulation while 76.6% showed true anestrus [13]. Unlike in cattle, not the uterine pathology but the ovarian acyclicity was found to be the major problem in anestrus as well as culled buffaloes [10]. Thus, it is understood that anestrus is the major form of infertility in Nepalese buffaloes that hinders their reproductive as well as productive efficiency [12,13,14,15].

## 4. Breeding Seasonality

In Nepal, environmental fluctuation is typical, from cold and semi-dry to a dry winter (December–February), rapidly increasing to a hot and dry spring (March–May), with a very hot and rainy monsoon summer (June–August) and moderate autumn (September–November) [10]. A distinct seasonal breeding pattern is prevalent in Nepalese buffaloes, with the late monsoon, autumn and early winter (i.e., July to December) as the best breeding seasons, and with spring and early summer (i.e., April to June) and January to March as the low and transitional breeding seasons, respectively [10,11]. As a consequence, the trend of parturition shows the maximum from June to December, with less parturition in January, April and May and almost no parturition in the months of February and March [10] (Figure 1). If we combine this problem with the summer infertility of dairy cows, it is apparent that the country suffers a severe drop in milk production during the spring and early summer months. 

While analyzing the year-round reproductive data of 226 anestrous buffaloes to examine the seasonal variations in the anestrus condition, it was found that the incidence of true anestrus with ovarian acyclicity was >70% during March to June, it peaked (>80%) in April/May and remained at around 50% in August and October-December. Moreover, a considerable proportion of anestrous buffaloes showed silent cyclicity throughout the year [11] (Figure 2). The incidence of anestrus in buffaloes is common during the summer months in India, as well as in other parts of the world [19,20]. Buffaloes suffer from the cessation of ovarian activity and silent heat during the months of dry spring and early summer. Most of the buffaloes exposed to extreme hot conditions cease ovarian activity [21,22]. 

Anestrus and low breeding activity during spring and early summer can be linked with the shortage of feed and fodder availability during these months. This might be the reason for the disrupted metabolic profile during this period, because the blood cholesterol level was reported to be significantly lower during the low breeding season than during the high breeding season [23]. Moreover, the extremely hot climatic conditions during these months may create a physiologically stressful condition for the buffaloes, causing acyclicity and anestrus. This assumption was supported by the finding that the cortisol level in the buffaloes was significantly higher during the low breeding season as compared to that during the high breeding season [23]. It is also important to link the seasonal pattern of buffalo reproduction to nutrition and fodder availability, which largely varies according to the rainfall pattern. Likewise, this seasonal breeding pattern of buffaloes can also be linked to photoperiodicity, because the breeding activity of buffaloes in the regions far from the equatorial zone is affected by the increasing length of the day [24,25,26]. 

## 5. Factors Associated with Anestrus and Compromising Its Treatment Response

Anestrus in Nepalese buffaloes was found to be associated mainly with endoparasitic infection [10,14], a poor nutritional status as reflected by a low body condition score (BCS) [11,14,27] and disrupted blood metabolic parameters [9,14,28]. In dairy animals, nutritional deficiency is one of the major causes of anestrus and other reproductive disorders. The process of follicular growth, maturation and ovulation is affected by the nutritional status of animals [29]. The nutritional status of animals can be subjectively assessed in terms of their BCS. BCS is a measure of the overall nutritional status, and, in particular, it indicates the energy dynamics of animals. It is an important factor influencing their reproductive performance [30]. In buffaloes, the phenomenon of a higher incidence of ovarian inactivity during dry spring and early summer is associated with a poor BCS, indicating a nutritional cause with poorly accessible nutrition during this time [11,31]. In Nepal, the proportion of true anestrus was reported to be significantly higher in poor-BCS buffaloes as compared to high-BCS buffaloes, suggesting that BCS is one of the major factors associated with true anestrus in buffaloes. Moreover, the blood calcium level was reported to be significantly lower in non-cyclic buffaloes than in cyclic buffaloes [28]. The blood cholesterol level was lower, while the blood cortisol level was higher, during the low breeding season in anestrous buffaloes [23], which indicated an insufficient roughage supply and increased stress in the buffaloes during this season. Likewise, the treatment response under synchronization protocols was low in buffaloes having low blood cholesterol and low protein levels, suggesting that these metabolic profiles are associated with anestrus and affect the success of treatment protocols [9]. 

In order to improve the fertility of anestrous buffaloes, various treatment approaches have been considered depending upon the type of anestrus, and a varying rate of success has been achieved. Hormonal treatments, particularly by administering PGF2 alpha to silently cycling buffaloes having a CL in the ovary and GnRH to silent buffaloes having a DF in the ovary, have shown a better response than other treatments, such as GnRH or vitamin mineral mixture (Vit-M) supplementation to acyclic buffaloes (Table 2).

It is obvious that treatment with PGF2 alpha produces a considerable response, with a consequent estrus and fertile ovulation in silently ovulated buffaloes bearing a CL in the ovary [10,13,32,33,34,35]. However, in the context of Nepal, low BCS before treatment negatively affected the pregnancy rate, and deficiencies in Ca and protein and gastrointestinal parasitic infection all compromised the nutritional statuses of animals and reduced the pregnancy rate after the treatment of anestrus buffaloes [14]. Therefore, producers should be informed that buffaloes with a low BCS may not respond to any form of treatment and should be encouraged to improve the nutritional statuses of the animals [9]. 

## 6. Technologies used to Enhance Reproduction 

### 6.1. Artificial Insemination 

In Nepal, the government started an artificial insemination (AI) project for buffaloes in 1985, implemented by the Animal Breeding Division, Khumaltar, Lalitpur. Later, in 2001, the Division was relocated to Lamapatan, Pokhara. In 2004, it was named the National Livestock Breeding Center, which was responsible for producing frozen semen from dairy animals. In buffaloes, only the Murrah bull’s frozen semen was produced from this center and was distributed across the country. Artificial insemination in buffaloes using frozen Murrah bull semen was introduced for the cross-breeding and genetic improvement of local and genetically inferior buffaloes. However, so far, the coverage of AI in buffaloes in Nepal is quite low; it was estimated to be 6.30% in the year 2020/2021 [36]. Thus, natural mating remains the predominant form of breeding in buffaloes in Nepal, which is common in the global context as well. Artificial insemination is more difficult in buffaloes compared with cattle due to the variable estrus cycles, reduced estrus behavior and reproductive seasonality. The latter is associated with a higher incidence of anestrus [11,20,37] and increased embryonic mortality [37] during the low breeding season. For the successful application of AI, it is important to understand the reproductive physiology, as well as the factors that are associated with compromised fertility. A number of hormone-based protocols for timed artificial insemination (TAI) have been developed in buffaloes; however, their application in Nepal is almost negligible. Such TAI protocols are ideal for buffaloes as they are known for their silent estrus behavior [11], and the clinical reproductive examination of the buffalo ovary is relatively difficult due to its smaller size, and the CLs in most cases are embedded in the ovary [38]. Past studies in Nepal focused on characterizing these infertility issues, and on understanding the seasonal breeding patterns and the factors causing anestrus and compromising its treatment response in buffaloes while treating them with specific hormones or applying different TAI protocols during different seasons. All the studies performed so far are limited to Murrah and their cross-bred populations, managed by small-scale farmers or government farms.

### 6.2. Application of Timed Artificial Insemination Protocols

Silent ovulation during the high breeding season and ovarian acyclicity during the transitional and low seasons are the major causes of anestrus in Nepalese buffaloes [11]. Progesterone-based protocols are useful in inducing ovarian cyclicity in acyclic anestrous buffaloes [39]. Similarly, it was established that GnRH-based protocols can be effectively applied in cyclic dairy animals. Considering these facts, most of the experiments in Nepal have been conducted with a 7-day CIDR co-synch protocol (D0: GnRH I and CIDR in, D7: CIDR out and PGF2 alpha treatment, D10 at around 60 h of PG treatment: fixed-time insemination and GnRH II) during the transitional and low breeding seasons and an ovsynch protocol (D0: GnRH I, D7: PGF2 alpha, D9: GnRH II and D10: fixed-time insemination after 16–20 h of GnRH II) during the high breeding season, although some experiments have been attempted using both protocols in all three seasons. There was a varying pregnancy outcome when applying these protocols (Table 3). Moreover, the effects of BCS on pregnancy outcomes was monitored, which indicated that BCS has an association, particularly during the low breeding season [9].

Recently, based on the association of endoparasitic infection and nutritional status with the treatment effect regarding the anestrous condition in buffaloes in Nepal, anthelmintic treatment at least one month prior to the initiation of the TAI protocol, and nutritional management by means of providing vitamin–mineral supplementation, in addition to improving the roughage supply soon after anthelmintic treatment until 2 weeks after AI, has been adopted. These interventions are termed integrated techniques of fertility management programs. The results indicated that the technique produces a more successful pregnancy, particularly during the low breeding season (Table 3). 

## 7. Gaps and Opportunities

There is broad potential to increase the productivity of buffaloes through the improvement of their reproductive performance. Anestrus is the most important cause of poor reproductive performance in buffaloes. It is also a major reproductive problem in modern dairy cow production worldwide. In commercial dairy farming, dairy herds are regularly visited by veterinarians, and cows with anestrus are treated with hormones for the synchronization of estrus and/or ovulation without delay. However, such regular reproductive examination of buffaloes at an appropriate interval is practically difficult in Nepal due to constraints such as the costs and availability of veterinary services in rural areas. For many marginal buffalo farms, infertility camps, which are organized once a year or at a longer interval, mainly by local livestock service offices, veterinary schools, dairy cooperatives and some other organizations, are the only opportunities for farmers to have their buffaloes examined and treated by veterinarians [14]. On the other hand, due to the limited number of field veterinarians available, most of the reproductive services are provided by veterinary technicians. They are the major extension workers who disseminate the latest technologies, such as the integrated techniques of fertility management programs, which are developed to be utilized in production farms. However, it is beyond their capacity to understand the nature and physiology underlying infertility problems and to provide an appropriate treatment or an appropriate tool generated from modern research. Therefore, it is imperative to ensure the availability of skilled veterinarians at a local level to meet the demands of modern buffalo reproduction management. The combined efforts of the governmental, and non-governmental organizations and educational and research institutions will be valuable to address the present gaps and create an opportunity to enhance the fertility and overall production efficiency of this important livestock.

## 8. Conclusions

Endoparasitic infection, a lower nutritional status and breeding seasonality are the major issues compromising the reproductive efficiency of buffaloes in Nepal. There is potential to enhance buffalo productivity in Nepal through the adoption of research-based fertility improvement technologies. The wider adoption of integrated techniques of fertility management as a package of practices could be key to improving the reproductive efficiency of buffaloes in Nepal.

## Figures and Tables

**Figure 1 animals-13-00070-f001:**
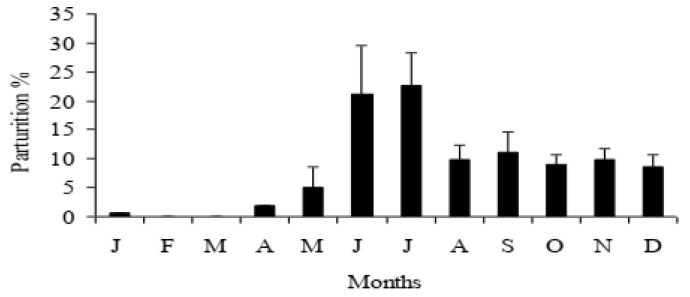
The annual trend of parturition of buffaloes in the Livestock Farm of the Institute of Agriculture and Animal Science, Tribhuvan University, Chitwan, Nepal [10].

**Figure 2 animals-13-00070-f002:**
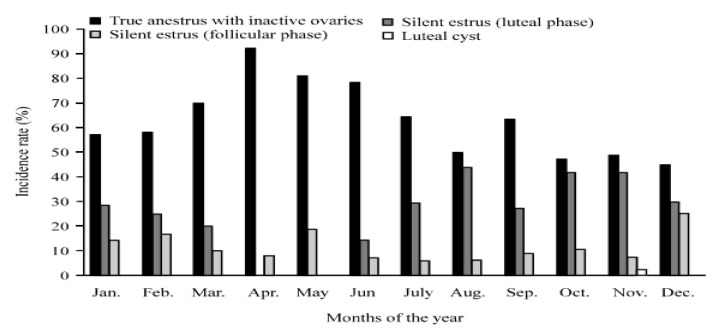
Monthly incidence of anestrus condition in buffaloes in Nepal [11].

**Table 1 animals-13-00070-t001:** Major reproductive parameters of indigenous buffalo breeds in Nepal.

Parameters (Mean ± SE)	Breed			
	Lime	Parkote	Gaddi	Terai
Age at first service (months)	51.6 ± 0.6	51.8± 0.55	45.6 ± 0.7	34.0 ± 0.6
Age at first calving (months)	61.2 ± 0.5	62.2 ± 1.6	68.4 ± 0.4	44.0 ± 0.1
Calving to conception interval (days)	190	175	350	NA
Calving interval (months)	21.0 ± 0.8	20.6 ± 1.0	23.4 ± 0.5	NA
Gestational length (days)	315.0 ± 1.7	315.0 ± 1.4	330.0 ± 1.4	305.0 ± 1.4
References	[17]	[17]	[18]	[7]

NA: Not available.

**Table 2 animals-13-00070-t002:** Responses to various treatments for anestrus and silent estrus in buffaloes.

Type of Anestrus	Treatment Used	No. of Buffaloes	Estrus Expression Rate (%)	Conception Rate (%)	References
True anestrus with ovarian acyclicity	Vit-M	76	64.5	82.9	[13]
	GnRH	7	NA	28.6	[14]
	Vit-M	24	NA	16.7	
	Phosphorus * injection	34	29.4	NA	[15]
Silent estrus in luteal phase	PGF 2 alpha	10	80.0	NA	[15]
	PGF 2 alpha	10	NA	60	[14]
	Vit-M	13	NA	23.1	
	PGF 2 alpha	22	100	100	[13]
	Vit-M	44	63.6	79.5	
Silent estrus in follicular phase	GnRH	18	55.6	NA	[15]
	GnRH	11	100	100	[13]
	Vit-M	81	64.2	81.5	
Anestrus with follicular cyst	GnRH	12	100	100	[13]

PGF2 alpha: Prostaglandin F2 alpha, GnRH: Gonadotrophin releasing hormone, Vit-M: Vitamin mineral mixture supplementation, * Sodium Salt of 4-dimethylamino-2-methylphenyl-phosphinic acid, NA: Not available.

**Table 3 animals-13-00070-t003:** Application of timed artificial insemination protocols along with integrated management program * in Nepalese buffaloes during different seasons.

Protocols	Season	Number of Animals	Estrus Expression (%)	Conception Rate (%)	References
Ovsynch	Good	14	50.0	64.3	[40]
Ovsynch	Good	35	54.3	45.7	[41]
CIDR co-synch	Transition	14	73.8	50.0	[41]
CIDR synch *	Low	14	100.0	42.9	[42]
CIDR co synch	Low	14	NA	21.4	[9]
CIDR co synch	Good	6	NA	66.7	[9]
Ovsynch	Good	15	NA	46.7	[43]
G6G	Good	7	NA	14.3	[43]
CIDR-PGF2 alpha	Low	30	81.0	26	[44]
Ovsynch *	Good	63	NA	55.5	[45]
CIDR co-synch *	Transition	45	NA	48.9	[45]
CIDR co-synch *	Low	99	NA	45.5	[45]
Ovsynch	Good	14	NA	42.9	[46]
CIDR co-synch	Low	14	NA	28.6	[46]
New CIDR co-synch	Low	19	78.9	10.5	[47]
Once used CIDR cosynch	Low	31	80.6	19.4	[47]

Ovsynch: Day 0 GnRH I, Day 7 PGF2 alpha, Day 9 GnRH II and Day 10 fixed-time insemination after 16–20 h of GnRH II; CIDR synch: Day 0 GnRH I CIDR in, Day 6 PGF2 alpha, Day 7 CIDR out, Day 9 GnRH II and Day 10 fixed-time insemination after 16–20 h of GnRH II; CIDR co-synch: Day 0 GnRH I and CIDR in, Day 7 CIDR out and PGF2 alpha treatment, Day 10 (around 60 h of PG treatment) fixed-time insemination and GnRH II; G6G: PGF2 alpha followed 2 days later with GnRH and then 6 days later followed by Ovsynch. *: Applied integrated technique of anthelmintic treatment at least one month prior to the initiation of protocol and nutritional management by means of improving roughage supply and administering Vit-M soon after anthelmintic treatment until minimum of 2 weeks after the timed artificial insemination. NA: Not available.

## Data Availability

Not applicable.

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
