# Peer review of "Reproduction and Fertility of Buffaloes in Nepal"

_animals, 2022, doi:10.3390/ani13010070_

Round 1
Reviewer 1 Report
Buffalo is the major livestock commodity in Nepal that contributes more than half of the total milk and more than one-third of the meat production in the country. The authors summarized the reproductive performance, infertility, seasonal breeding, factors associated with anestrus, and compromising its treatment response, as well as the reproductive technologies application in the Buffalo industry in Nepal. In addition, the authors also look forward to the shortcomings and future development opportunities of the buffalo industry in Nepal. It is a good review and can provide help in understanding the current status of the buffalo industry in Nepal and determining future research directions. I recommended it for publication in this journal. Aside from the fact that English needs polishing, there are no other suggestions.
Author Response
Point 1: Buffalo is the major livestock commodity in Nepal that contributes more than half of the total milk and more than one-third of the meat production in the country. The authors summarized the reproductive performance, infertility, seasonal breeding, factors associated with anestrus, and compromising its treatment response, as well as the reproductive technologies application in the Buffalo industry in Nepal. In addition, the authors also look forward to the shortcomings and future development opportunities of the buffalo industry in Nepal. It is a good review and can provide help in understanding the current status of the buffalo industry in Nepal and determining future research directions. I recommended it for publication in this journal. Aside from the fact that English needs polishing, there are no other suggestions.
Response 1: The authors are grateful to the reviewer for recommending the manuscript for publication in this journal. We have heartily accepted the suggestion of the reviewer of polishing the English language.
Reviewer 2 Report
General comments:
The manuscript entitled "Reproduction and fertility of buffaloes in Nepal” (Manuscript number; animals-2089409) was describes. This manuscript is very interesting and provides useful information for the basic information of reproductive characteristics and applied reproductive technology in buffalos in Nepal. There are some points needing corrections in the manuscript. Please consider the suggested edits listed below.
Specific Comments:
2. Reproductive perforemance or 3. Infertility
Please show the prevalence rates of anestrus buffalo in total and each month or “the low breeding season” and “the good breeding season”. Readers want to know the situation of the ratio of estrus and anestrus buffalo in Nepal field.
Line 146-147
Please clarify the months of “the low breeding season” and “the good breeding season”, respectively.
Author Response
Point 1: The manuscript entitled "Reproduction and fertility of buffaloes in Nepal” (Manuscript number; animals-2089409) was describes. This manuscript is very interesting and provides useful information for the basic information of reproductive characteristics and applied reproductive technology in buffalos in Nepal. There are some points needing corrections in the manuscript. Please consider the suggested edits listed below.
Response 1: The authors are grateful to the reviewer.
Specific Comments:
Point 2: Reproductive performance or 3. Infertility
Please show the prevalence rates of anestrus buffalo in total and each month or “the low breeding season” and “the good breeding season”. Readers want to know the situation of the ratio of estrus and anestrus buffalo in Nepal field.
Response 2: Thank you very much. There are no data available in Nepal regarding the prevalence rates of anestrus buffalo in total during different seasons. However, we have shown the proportions of types of anestrus under the sub-heading "Infertility" (lines 110 to 120). Similarly, we have also shown the monthly incidence of anestrus conditions under the sub-heading "Breeding seasonality" (lines 137-142) that is also shown in Fig. 2.
Point 3: Line 146-147
Please clarify the months of “the low breeding season” and “the good breeding season”, respectively.
Response 3: Thank you very much. We have clearly mentioned the months of breeding seasons under the sub-heading "Breeding seasonality" (lines 125-128). For clarification, a distinct seasonal breeding pattern is prevalent in Nepalese buffaloes with the late monsoon, autumn and early winter (i.e. July to December) as the good breeding season while spring and early summer (i.e. April to June) as the low and January to March as the transition breeding seasons.
Reviewer 3 Report
Dear authors, I have evaluated your review article identified as animals-2089409. In my opinion, the manuscript examines in a correct and articulated way the issue of reproduction efficiency and fertility of buffalo reared in Nepal. I find the manuscript well-constructed, with logically articulated and adequately exposed contents. The scientific soundness is remarkable, and the literature cited is, overall, functional to the review's purposes. In this regard, however, I suggest expanding the literature cited with a recent review (https://doi.org/10.3390/ani11092683) that addresses, among others, the issue of poor reproduction efficiency in draught buffaloes. As far as I know, Nepal provides an excellent example of the role of draught animals in supporting rural labor in small-scale mixed farms. In this context, buffaloes are commonly used in land preparation, especially on the flat plains in the country's south. For this class of animals, the poor fertility induced by seasonality, parasitic, and feeding issues could be exacerbated by their use as power sources. This aspect is underlined in a cited literature review, that the authors could use as a reference to address the fertility problems associated with the use of buffalo as a working animal, which in my view is peculiar to Nepali agriculture. I believe that the authors could find space in the introduction for this aspect, alluding to the importance of the buffalo, as well as to produce milk and meat, as a draft animal and to the reproduction problems associated with this use.
My other suggestions, listed below, are related to formatting and editing issues:
- throughout the manuscript, I noticed an excess of spacing (e.g., on lines 26, 89. etc.). Authors are invited to check. Thanks.
- some sentences (for example, lines 106, 155, etc.) do not end with a period. Authors are invited to check. Thanks.
- L 98 (Table 1): in my opinion, the references could be better reported in a specific column (as done for table 2) rather than as footnotes. Thanks.
- L 130 and 142: as per the journal's template, the titles of the tables should be aligned with the body of the text. Thanks.
- L 184 (Table 2): authors are invited to check the table (indents, borders, etc.) and the footnotes (line spacing, character, etc.) formatting. Thanks.
Author Response
Point 1. Dear authors, I have evaluated your review article identified as animals-2089409. In my opinion, the manuscript examines in a correct and articulated way the issue of reproduction efficiency and fertility of buffalo reared in Nepal. I find the manuscript well-constructed, with logically articulated and adequately exposed contents. The scientific soundness is remarkable, and the literature cited is, overall, functional to the review's purposes. In this regard, however, I suggest expanding the literature cited with a recent review (https://doi.org/10.3390/ani11092683) that addresses, among others, the issue of poor reproduction efficiency in draught buffaloes. As far as I know, Nepal provides an excellent example of the role of draught animals in supporting rural labor in small-scale mixed farms. In this context, buffaloes are commonly used in land preparation, especially on the flat plains in the country's south. For this class of animals, the poor fertility induced by seasonality, parasitic, and feeding issues could be exacerbated by their use as power sources. This aspect is underlined in a cited literature review, that the authors could use as a reference to address the fertility problems associated with the use of buffalo as a working animal, which in my view is peculiar to Nepali agriculture. I believe that the authors could find space in the introduction for this aspect, alluding to the importance of the buffalo, as well as to produce milk and meat, as a draft animal and to the reproduction problems associated with this use.
Response 1: The authors are grateful to the reviewer. We have added in the sub-heading "Introduction" (lines 44-49) as, in addition to milk and meat production, the buffalo also has significant contribution to the farm economy through draught power and farm yard manure [Mota-Rojas et al., 2021]. In Nepal, the male buffaloes especially on the flat plains in the country's south, also contribute as a draught power mainly in land preparation and transportation of farm products in smallholder farms. However, female buffaloes are not used for draught power. Therefore, its association to infertility is not reported in Nepal and we could not mention this point in the manuscript.
My other suggestions, listed below, are related to formatting and editing issues:
Point 2.- throughout the manuscript, I noticed an excess of spacing (e.g., on lines 26, 89. etc.). Authors are invited to check. Thanks.
Response 2: Thank you very much. We have thoroughly re-checked the manuscript and made corrections.
Point 3.- some sentences (for example, lines 106, 155, etc.) do not end with a period. Authors are invited to check. Thanks.
Response 3: Thank you very much. We have made corrections.
Point 4.- L 98 (Table 1): in my opinion, the references could be better reported in a specific column (as done for table 2) rather than as footnotes. Thanks.
Response 4: Thank you very much. In Table 1, the references are mentioned in the specific column as done in Table 2.
Point 5.- L 130 and 142: as per the journal's template, the titles of the tables should be aligned with the body of the text. Thanks.
Response 5: Thank you very much. We have done it for all the tables.
Point 6.- L 184 (Table 2): authors are invited to check the table (indents, borders, etc.) and the footnotes (line spacing, character, etc.) formatting. Thanks.
Response 6: Thank you very much. We have done it for all the tables.